# *Slice-based Learning*: A Programming Model for Residual Learning in Critical Data Slices

**Vincent S. Chen, Sen Wu, Zhenzhen Weng, Alexander Ratner, Christopher Ré**

Stanford University
vincentsc@cs.stanford.edu, senwu@stanford.edu, zzweng@stanford.edu,
ajratner@stanford.edu, chrismre@cs.stanford.edu

## Abstract

In real-world machine learning applications, data subsets correspond to especially critical outcomes: vulnerable cyclist detections are safety-critical in an autonomous driving task, and "question" sentences might be important to a dialogue agent's language understanding for product purposes. While machine learning models can achieve high quality performance on coarse-grained metrics like F1-score and overall accuracy, they may underperform on critical subsets—we define these as *slices*, the key abstraction in our approach. To address slice-level performance, practitioners often train separate "expert" models on slice subsets or use multi-task hard parameter sharing. We propose *Slice-based Learning*, a new programming model in which the *slicing function (SF)*, a programming interface, specifies critical data subsets for which the model should commit additional capacity. Any model can leverage SFs to learn *slice expert representations*, which are combined with an attention mechanism to make *slice-aware* predictions. We show that our approach maintains a parameter-efficient representation while improving over baselines by up to 19.0 F1 on slices and 4.6 F1 overall on datasets spanning language understanding (e.g. SuperGLUE), computer vision, and production-scale industrial systems.

## 1 Introduction

In real-world applications, some model outcomes are more important than others: for example, a data subset might correspond to safety-critical but rare scenarios in an autonomous driving setting (e.g. detecting cyclists or trolley cars [19]) or critical but lower-frequency healthcare demographics (e.g. bone X-rays associated with degenerative joint disease [27]). Traditional machine learning systems optimize for overall quality, which may be too coarse-grained; models that achieve high overall performance might produce unacceptable failure rates on *slices* of the data. In many production settings, the key challenge is to maintain overall model quality while improving slice-specific metrics.

To formalize this challenge, we introduce the notion of *slices*: application-critical data subsets, specified programmatically by machine learning practitioners, for which we would like to improve model performance. This leads to three technical challenges:

- **Coping with Noise:** Defining slices precisely can be challenging. While engineers often have a clear intuition of a slice, typically as a result of an error analysis, translating that intuition into a machine-understandable description can be a challenging problem, e.g., *"the slice of data that contains a yellow light at dusk."* As a result, any method must be able to cope with imperfect, overlapping definitions of data slices, as specified by noisy or *weak supervision*.

- **Stable Improvement of the Model:** Given a description of a set of slices, we want to improve the prediction quality on each of the slices without hurting overall model performance. Often,

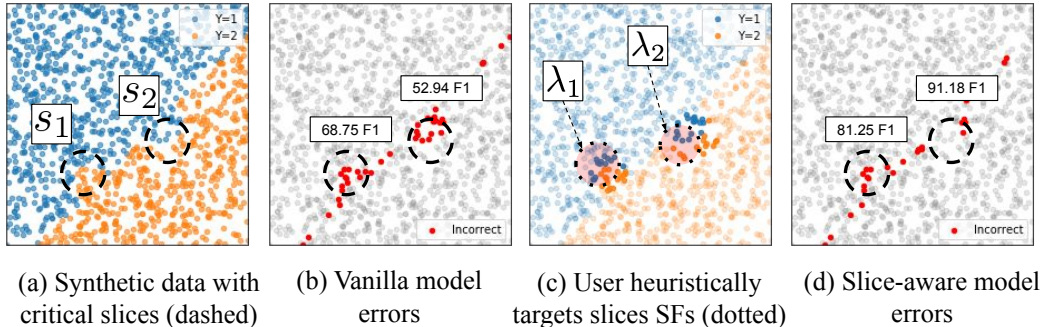

| (a) Synthetic data with critical slices (dashed) | (b) Vanilla model errors | (c) User heuristically targets slices SFs (dotted) | (d) Slice-aware model errors |

Figure 1: ***Slice-based Learning* via synthetically generated data:** (a) The data distribution contains critical slices, $s_1$, $s_2$, that represent a small proportion of the dataset. (b) A vanilla neural network correctly learns the general, linear decision boundary but fails to learn the perturbed slice boundary. (c) A user writes *slicing functions* (SFs), $\lambda_1$, $\lambda_2$, to heuristically target critical subsets. (d) The model commits additional capacity to learn slice expert representations. Upon reweighting slice expert representations, the slice-aware model learns to classify the fine-grained slices with higher F1 score.

these goals are in tension: in many baseline approaches, steps to improve the slice-specific model performance would degrade the overall model performance, and vice-versa.

- **Scalability:** There may be many slices. Indeed, in industrial deployments of slice-based approaches, hundreds of slices are commonly introduced by engineers [32]—any approach to *Slice-based Learning* must be judicious with adding parameters as the number of slices grow.

To improve *fine-grained*, i.e. slice-specific, performance, an intuitive solution is to create a separate model for each slice. To produce a single prediction at test time, one often trains a *mixture of experts* model (MoE) [18]. However, with the growing size of ML models, MoE is often untenable due to runtime performance, as it could require training and deploying hundreds of large models—one for each slice. Another strategy draws from multi-task learning (MTL), in which slice-specific *task heads* are learned with hard-parameter sharing [7]. This approach is computationally efficient but may not effectively share training data across slices, leading to suboptimal performance. Moreover, in MTL, tasks are distinct, while in *Slice-based Learning*, a single *base task* is refined by related slice tasks.

We propose a novel programming model, called *Slice-based Learning*, in which practitioners provide slicing functions (SFs), a programming abstraction for heuristically targeting data subsets of interest. SFs coarsely map input data to slice indicators, which specify data subsets for which we should allocate additional model capacity. To improve slice-level performance, we introduce *slice-residual-attention modules* (SRAMs) that explicitly model *residuals* between slice-level and the overall task predictions. SRAMs are agnostic to the architecture of any neural network model that they are added to—which we refer to as the *backbone* model—and we demonstrate our approach on state-of-the-art text and image models. Using shared backbone parameters, our model initializes slice "expert" representations, which are associated with learning slice-membership indicators and class predictors for examples in a particular slice. Then, slice indicators and prediction confidences are used in an *attention-mechanism* to reweight and combine each slice expert representation based on learned residuals from the base representation. This produces a *slice-aware* featurization of the data, which can be used to make a final prediction.

Our work fits into an emerging class of programming models that sit on top of deep learning systems [19, 30]. We are the first to introduce and formalize *Slice-based Learning*, a key programming abstraction for improving ML models in real-world applications subject to slice-specific performance objectives. Using an independent error analysis for the recent GLUE natural language understanding benchmark tasks [39], by simply encoding the identified error categories as slices in our framework, we show that we can improve the quality of state-of-the-art models by up to 4.6 F1 points, and we observe slice-specific improvements of up to 19.0 points. We also evaluate our system on autonomous vehicle data and show improvements up to 15.6 F1 points on context-dependent slices (e.g., presence of bus, traffic light) and 2.3 F1 points overall. Anecdotally, when deployed in production systems [32], *Slice-based Learning* provides a practical programming model with improvements of up to 40 F1 points in critical test-time slices. On the SuperGlue benchmark [38], this procedure accounts for a 2.7

improvement in aggregate benchmark score using the same architecture as previous state-of-the-art submissions. In addition to the proposal of SRAMs, we perform an in-depth analysis to explain the mechanisms by which SRAMs improve quality. We validate the efficacy of quality and noise estimation in SRAMs and compare to weak supervision frameworks [30] that estimate the quality of supervision sources to improve overall model accuracy. We show that by using SRAMs, we are able to produce accurate quality estimates, which leads to higher downstream performance on such tasks by an average of 1.1 overall F1 points.

## 2 Related Work

Our work draws inspiration from three main areas: mixture of experts, multi-task learning, and weak supervision. Jacobs et. al [18] proposed a technique called **mixture of experts** that divides the data space into different homogeneous regions, learns the regions of data separately, and then combines results with a single gating network [37]. This work is a generalization of popular ensemble methods, which have been shown to improve predictive power by reducing overfitting, avoiding local optima, and combining representations to achieve optimal hypotheses [36]. We were motivated in part by reducing the runtime cost and parameter count for such models.

**Multi-task learning** (MTL) models provide the flexibility of *modular* learning—specific task heads, layers, and representations can be changed in an application-specific, ad hoc manner. Furthermore, MTL models benefit from the computational efficiency and regularization afforded by hard parameter sharing [7]. There are often also performance gains seen from adding auxiliary tasks to improve representation learning objectives [8, 33]. While our approach draws high-level inspiration from MTL, we highlight key differences: whereas tasks are disjoint in MTL, slice tasks are formulated as *micro-tasks* that are direct extensions of a base task—they are designed specifically to learn deviations from the base-task representation. In particular, sharing information, as seen in cross-stitch networks [26], requires $\Omega(n^2)$ weights across $n$ local tasks; our formulation only requires attention over $O(n)$ weights, as slice tasks operate on the *same* base task. For example, practitioners might specify yellow lights and night-time images as important slices; the model learns a series of micro-tasks—based solely on the data specification—to inform how its approach for the base task, object detection, should change in these settings. As a result, slice tasks are not fixed ahead of time by an MTL specification; instead, these micro-task boundaries are learned dynamically from corresponding data subsets. This style of information sharing sits adjacent to cross-task knowledge literature in recent MTL models [35, 42], and we were inspired by these methods.

**Weak supervision** has been viewed as a new way to incorporate data of varying accuracy sources, including domain experts, crowd sourcing, data augmentations, and external knowledge bases [2, 5, 6, 11, 14, 21, 25, 29, 31]. We take inspiration from *labeling functions* [31] in weak supervision as a programming paradigm, which has seen success in industrial deployments [2]. In existing weak supervision literature, a key challenge is to assess the accuracy of a training data point, which is a function of supervision sources. In this work, we model this accuracy using learned representations of user-defined slices—this leads to higher overall quality.

Weak supervision and multitask learning can be viewed as orthogonal to slicing: we have observed them used alongside *Slice-based Learning* in academic projects and industrial deployments [32].

## 3 Slice-based Learning

We propose *Slice-based Learning* as a programming model for training machine learning models where users specify important data subsets to improve model performance. We describe the core technical challenges that lead to our notion of *slice-residual-attention modules* (SRAMs).

### 3.1 Problem statement

To formalize the key challenges of slice-based learning, we introduce some basic terminology. In our *base task*, we use a supervised input, $(x \in \mathcal{X}, y \in \mathcal{Y})$, where the goal is to learn according to a standard loss function. In addition, the user provides a set of $k$ functions called *slicing functions (SFs)*, $\{\lambda_1, \ldots, \lambda_k\}$, in which $\lambda_i : \mathcal{X} \to \{0, 1\}$. These SFs are not assumed to be perfectly accurate; for example, SFs may be based on noisy or *weak* supervision sources in functional form [31]. SFs can

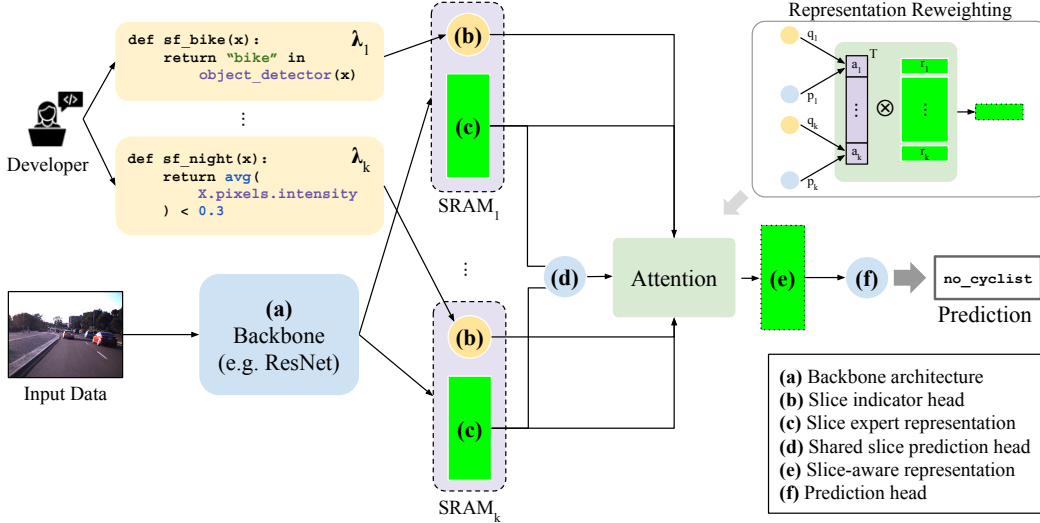

Figure 2: **Model Architecture**: A developer writes SFs ($\lambda_{i=1,\ldots,k}$) over input data and specifies any **(a) backbone** architecture (e.g. ResNet [16], BERT [13]) as a feature extractor. These features are shared parameters for $k$ slice-residual attention modules (SRAMs); each learns a **(b) slice indicator head** which outputs a prediction, $q_i$, indicating which slice the example belongs to, as supervised by $\lambda_i$. SRAMs also learn a **(c) slice expert representation**, trained only on examples belonging to the slice using a **(d) shared slice prediction head**, which makes predictions, $p_i$, on the original task schema and is supervised by the masked ground truth labels for the corresponding slice. An attention mechanism, $a$, reweights these representations, $r_i$, into a combined, **(e) slice-aware representation**. A final **(f) prediction head** makes model predictions based on this slice-aware representation.

come from domain-specific heuristics, distant supervision sources, or other off-the-shelf models, as seen in Figure 2. Ultimately, the model's goal is to improve (or avoid damaging) the overall accuracy on the base task while improving the model on the specified slices.

Formally, each of $k$ slices, denoted $s_{i=1,\ldots,k}$, is an unobserved, indicator random variable, and each user-specified SF, $\lambda_{i=1,\ldots,k}$ is a corresponding, noisy specification. Given an input tuple $(\mathcal{X}, \mathcal{Y}, \{\lambda_i\}_{i=1,\ldots,k})$ consisting of a dataset $(\mathcal{X}, \mathcal{Y})$, and $k$ different user-defined SFs $\lambda_i$, our goal is to learn a model $f_{\hat{w}}(\cdot)$—i.e. estimate model parameters $\hat{w}$—that predicts $P(Y|\{s_i\}_{i=1,\ldots,k}, \mathcal{X})$ with high slice-specific accuracies without substantially degrading overall accuracy.

**Example 1** *A developer notices that their self-driving car is not detecting cyclists at night. Upon error analysis, they diagnose that their state-of-the-art object detection model, trained on an automobile detection dataset $(\mathcal{X}, \mathcal{Y})$ of images, is indeed underperforming on night and cyclist slices. They write two SFs: $\lambda_1$ to classify night vs. day, based on pixel intensity; and $\lambda_2$ to detect bicycles, which calls a pretrained object detector for a bicycle (with or without a rider). Given these SFs, the developer leverages Slice-based Learning to improve model performance on safety-critical subsets.*

Our problem setup makes a key assumption: SFs may be *non-servable* during test-time—i.e, during inference, an SF may be unavailable because it is too expensive to compute or relies on private metadata [1]. In Example 1, the potentially expensive cyclist detection algorithm is non-servable at runtime. When our model is served at inference, *SFs are not necessary*, and we can rely on the model's *learned* indicators.

## 3.2 Model Architecture

The *Slice-based Learning* architecture has six components. The key intuition is that we will train a standard prediction model, which we call the *base task*. We then learn a representation for each slice that explains how its predictions should differ from the representation of the base task—i.e., a *residual*. An attention mechanism then combines these representations to make a slice-aware prediction.

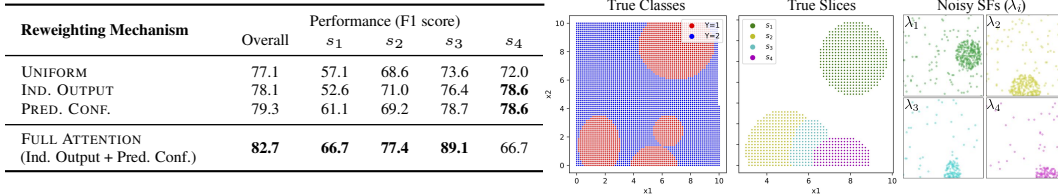

| Reweighting Mechanism | Performance (F1 score) | | | | |
|---|---|---|---|---|---|
| | Overall | $s_1$ | $s_2$ | $s_3$ | $s_4$ |
| UNIFORM | 77.1 | 57.1 | 68.6 | 73.6 | 72.0 |
| IND. OUTPUT | 78.1 | 52.6 | 71.0 | 76.4 | **78.6** |
| PRED. CONF. | 79.3 | 61.1 | 69.2 | 78.7 | **78.6** |
| FULL ATTENTION (Ind. Output + Pred. Conf.) | **82.7** | **66.7** | **77.4** | **89.1** | 66.7 |

Figure 3: **Architecture Ablation**: Using a synthetic, two-class dataset (Figure, left) with four randomly specified (size, shape, location) slices (Figure, middle), we specify corresponding, noisy SFs (Figure, right) and ablate specific model components by modifying the reweighting mechanism for slice expert representations. We compare overall/slice performance for uniform, indicator output, prediction confidence weighting, and the proposed attention weighting using all components. Our FULL ATTENTION approach performs most consistently on slices without worsening overall performance.

With this intuition in mind, the six components (Figure 2) are: (a) a **backbone**, (b) a set of $k$ **slice-indicator heads**, and (c) $k$ corresponding **slice expert representations**, (d) a **shared slice prediction head**, (e) a combined, **slice-aware representation**, and (f) a **prediction head**. Each SRAM operates over any backbone architecture and represents a path through components (b) through (e). We describe the architecture assuming a binary classification task (output dim. $c = 1$):

**(a) Backbone:** Our approach is agnostic to the neural network architecture, which we call the *backbone*, denoted $f_{\hat{w}}$, which is used primarily for feature extraction (e.g. the latest transformer for textual data, CNN for image data). The backbone maps data points $x$ to a representation $z \in \mathbb{R}^d$.

**(b) Slice indicator heads:** For each slice, an indicator head will output an input's slice membership. The model will later use this to reweight the "expert" slice representations based on the likelihood that an example is in the corresponding slice. Each indicator head maps the backbone representation, $z$, to a logit indicating slice-membership: $\{q_i\}_{i=1,\dots,k} \in \mathbb{R}$. Each slice indicator head is supervised by the output of a corresponding SF, $\lambda_i$. For each example, we minimize the multi-label binary cross entropy loss ($\mathcal{L}_{CE}$) between the unnormalized logit output of each $q_i$ and $\lambda_i$: $\ell_{ind} = \sum_i^k \mathcal{L}_{CE}(q_i, \lambda_i)$

**(c) Slice expert representations:** Each slice representation, $\{r_i\}_{i=1,\dots,k}$, will be treated as an "expert" feature for a given slice. We learn a linear mapping from the backbone, $z$, to each $r_i \in \mathbb{R}^h$, where $d'$ is the size of all slice expert representations.

**(d) Shared slice prediction head:** A shared, slice prediction head, $g(\cdot)$, maps each slice expert representation, $r_i$, to a logit, $\{p_i\}_{i=1,\dots,k}$, in the output space of the base task: $g(r_i) = p_i \in \mathbb{R}^c$, where $c = 1$ for binary classification. We train slice "expert" tasks using *only* examples belonging to the corresponding slice, as specified by $\lambda_i$. Because parameters in $g(\cdot)$ are shared, each representation, $r_i$, is forced to *specialize to the examples belonging to the slice*. We use the base task's ground truth label, $y$, to train this head with binary cross entropy loss: $\ell_{pred} = \sum_i^k \lambda_i \mathcal{L}_{CE}(p_i, y)$

**(e) Slice-aware representation:** For each example, the slice-aware representation is the combination of several "expert" slice representations according to 1) the likelihood that the input is in the slice and 2) the confidence of the slice "expert's" prediction. To explicitly model the residual from slice representations to the base representation, we initialize a trivial "base slice" which consists of *all examples* so that we have the corresponding indicators, $q_{BASE}$, and predictors, $p_{BASE}$.

Let $Q = \{q_1, \dots, q_k, q_{BASE}\} \in \mathbb{R}^{k+1}$ be the vector of concatenated slice indicator logits, $P = \{p_1, \dots, p_k, p_{BASE}\} \in \mathbb{R}^{c \times k+1}$ be the vector of concatenated slice prediction logits, and $R = \{r_1, \dots, r_k, r_{BASE}\} \in \mathbb{R}^{h \times k+1}$ be the $k + 1$ stacked slice expert representations. We compute our attention by combining the likelihood of slice membership, $Q$, and the slice prediction confidence, which we interpret as a function of the logits—in the binary case $c = 1$, we use $abs(P)$ as this confidence. We then apply a Softmax to create soft attention weights over the $k + 1$ slice expert

| Method | Performance (F1 score) | | |
|---|---|---|---|
| | Overall | $S_1$ | $S_2$ |
| VANILLA | 96.56 | 52.94 | 68.75 |
| DP [31] | 96.88 | 44.12 | 43.75 |
| HPS [7] | 96.72 | 50.00 | 75.00 |
| MOE [18] | **98.48** | 88.24 | **87.50** |
| SBL | 97.92 | **91.18** | 81.25 |

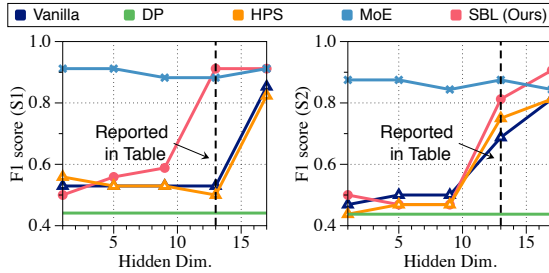

Figure 4: **Scaling with hidden feature representation dimensions.** We plot model quality versus the hidden dimension size. The slice-aware model (SBL) improves over *hard parameter sharing* (HPS) on both slices at a fixed hidden dimension size, while being close to *mixture of experts* (MOE). Note: MOE has significantly more parameters overall, as it copies the entire model.

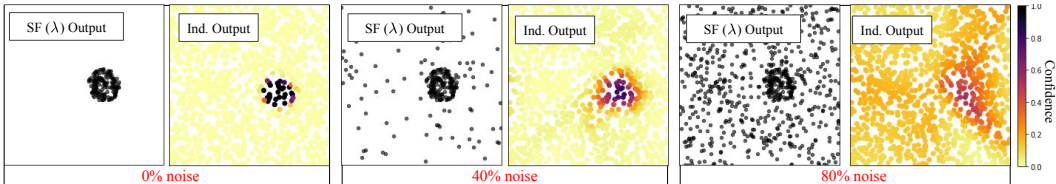

Figure 5: **Coping with Noise**: We test the robustness of our approach on a simple synthetic example. In each panel, we show noisy SFs (left) as binary points and the corresponding slice indicator's output (right) as a heatmap of probabilities. We show that the indicator assigns low relative probabilities on noisy (40%, middle) samples and ignores a very noisy (80%, right) SF, assigning relatively uniform scores to all samples.

representations: $a \in \mathbb{R}^{k+1} = \text{Softmax}(Q + abs(P))$. Using a weighted sum, we then compute the combined, slice-aware representation: $z' \in \mathbb{R}^{d'} = Ra$.

**(f) Prediction head**  Finally, we use our slice-aware representation $z'$ as the input to a final linear layer, $h(\cdot)$, which we term the *prediction head*, to make a prediction on the original, base task. During inference, this prediction head makes the final prediction. To train the prediction head, we minimize the cross entropy between the prediction head's output, $h(z')$, and the base task's ground truth labels, $y$: $\ell_{\text{base}} = \mathcal{L}_{\text{CE}}(h(z'), y)$.

Overall, the model is trained using loss values from all task heads: $\ell_{\text{train}} = \ell_{\text{base}} + \ell_{\text{ind}} + \ell_{\text{pred}}$. In Figure 3, we show ablations of this architecture in a synthetic experiment varying the components in the reweighting mechanism—specifically, our described attention approach outperforms using *only* indicator outputs, *only* predictor confidences, or uniform weights to reweight the slice representations.

### 3.3  Synthetic data experiments

To understand the properties of *Slice-based Learning* (SBL), we validate our model and its components (Figure 2) on a set of synthetic data. In the results demonstrated in Figure 1, we construct a dataset $\mathcal{X} \in \mathbb{R}^2$ with a 2-way classification problem in which over 95% of the data are linearly separable. We introduce two minor perturbations along the decision boundary, which we define as critical slices, $s_1$ and $s_2$. Intuitively, examples that fall within these slices follow different distributions ($P(\mathcal{Y}|\mathcal{X}, s_i)$) relative to the overall data ($P(\mathcal{Y}|\mathcal{X})$). For all models, the shared backbone is defined as a 2-layer MLP architecture with a backbone representation size $d = 13$ and a final $ReLU$ non-linearity. In SBL, the slice-expert representation is initialized with the same size: $d' = 13$.

**The model learns the slice-conditional label distribution** $P(Y|s_i, X)$ **from noisy SF inputs.** We show in Figure 1b that the slices at the perturbed decision boundary cannot be learned in the general case, by a VANILLA model. As a result, we define two SFs, $\lambda_1$ and $\lambda_2$, to target the slices of interest. Because our attention-based model (SBL) is slice-aware, it outperforms VANILLA, which

has no notion of slices (Figure 1d). Intuitively, if the model knows "where" in the 2-dim data space an example lives (as defined by SFs), it can condition on slice-specific features as it makes a final, slice-aware prediction. In Figure 5, we observe our model's ability to cope with noisy SF inputs: the indicator is robust to moderate amounts of noise by ignoring noisy examples (middle); with extremely noisy inputs, it disregards poorly-defined SFs by assigning relatively uniform weights (right).

**Overall model performance does not degrade.**    The primary goal of the slice-aware model is to improve slice-specific performance without degrading the model's existing capabilities. We show that SBL improves the overall score by 1.36 F1 points by learning the proportionally smaller perturbations in the decision boundary in addition to the more general linear boundary (Figure 4, left). Further, we note that we do not regress performance on individual slices.

**Learning slice weights with features $P(Y|s_i, X)$ improves over doing so with only supervision source information $P(Y|s_i)$.**    A core assumption of our approach asserts that if the model learns improved slice-conditional weights via $\lambda_i$, downstream slice-specific performance will improve. Data programming (DP) [31] is a popular weak supervision approach deployed at numerous Fortune 500 companies [2, 32], in which the weights of heuristics are learned solely from labeling source information. We emphasize that our setting provides the model with strictly more information—in the data's feature representations—to learn such weights; we show in Figure 4 (right) that increasing representation size allows us to significantly outperform DP.

**Attention weights learn from noisy $\lambda_i$ to combine slice residual representations.**    SBL achieves improvements over methods that do not aggregate slice information, as defined by each noisy $\lambda_i$. Both the indicator outputs ($Q$) and prediction confidence ($abs(P)$) are robustly combined in the attention mechanism. Even a noisy indicator will be upweighted if the predictions are high confidence, and if the indicator has high signal, even a slice expert making poor predictions can benefit from underlying slice-specific features. We show in Figure 4 that our method improves over HPS, which is slice-aware, but has no way of combining slice information despite increasingly noisy $\lambda_i$. In contrast, our attention-based architecture is able to combine slice expert representations, as SBL sees improvements over VANILLA by 38.2 slice-level F1 averaged across $s_1$ and $s_2$.

**SBL demonstrates similar expressivity to MoE with much less cost.**    With approximately half as many parameters, SBL comes within 6.25 slice-level F1 averaged across $s_1$ and $s_2$ of MoE (Figure 4). With large backbone architectures, characterized by $M$ parameters, and a large number of slices, $k$, MoE requires a quadratically large number of parameters, because we initialize an entire backbone for each slice. In contrast, all other models scale linearly in parameters with $M$.

## 4    Experiments

Compared to baselines using the same backbone architecture, we demonstrate that our approach successfully models slice importance and improves slice-level performance without impacting overall model performance. Then, we demonstrate our method's advantages in aggregating noisy heuristics, compared to existing weak supervision literature. We perform all empirical experiments on Google's Cloud infrastructure using NVIDIA V100 GPUs.

### 4.1    Applications

Using natural language understanding (NLU) and computer vision (CV) datasets, we compare our method to baselines commonly used in practice or the literature to address slice-specific performance.

#### 4.1.1    Baselines

For each baseline, we first train the backbone parameters with a standard hyperparameter search over learning rate and $\ell_2$ regularization values. Then, each method is initialized from the backbone weights and fine-tuned for a fixed number of epochs and the optimal hyperparameters.

VANILLA: A vanilla neural network backbone is trained with a final prediction head to make predictions. This baseline represents the de-facto approach used in deep learning modeling tasks; it is unaware of slices information and neglects to model them as a result.

| Dataset | CoLA (Matthews Corr. [24]) | | | | RTE (F1 Score) | | | | CyDet (F1 Score) | | | |
|---|---|---|---|---|---|---|---|---|---|---|---|---|
| | Param Inc. | Overall (std) | Slice Lift Max | Avg | Param Inc. | Overall (std) | Slice Lift Max | Avg | Param Inc. | Overall (std) | Slice Lift Max | Avg |
| Vanilla | – | 57.8 ($\pm$1.3) | – | – | – | 67.0 ($\pm$1.6) | – | – | – | 39.4 ($\pm$-5.4) | – | – |
| HPS [7] | 12% | 57.4 ($\pm$2.1) | +12.7 | 1.1 | 10% | 67.9 ($\pm$1.8) | +12.7 | +2.9 | 10% | 37.4 ($\pm$3.6) | +6.3 | -0.7 |
| Manual | 12% | 57.9 ($\pm$1.2) | +6.3 | +0.4 | 10% | 69.4 ($\pm$1.8) | +10.7 | +4.2 | 10% | 36.9 ($\pm$4.2) | +6.3 | -1.7 |
| MoE [18] | 100% | 57.2 ($\pm$0.9) | +20.0 | +1.3 | 100% | 69.2 ($\pm$1.5) | +10.9 | +3.9 | 100% | OOM | OOM | OOM |
| SBL | 12% | 58.3 ($\pm$0.7) | +19.0 | +2.5 | 10% | 69.5 ($\pm$0.8) | +10.9 | +4.6 | 10% | 40.9 ($\pm$3.9) | +15.6 | +2.3 |

Table 1: **Application Datasets**: We compare our model to baselines averaged over 5 runs with different seeds in natural language understanding and computer vision applications and note the relative increase in number of params for each method. We report the overall score and maximum relative improvement (denoted *Lift*) over the Vanilla model for each of the slice-aware baselines. For some trials of MoE, our system ran out of GPU memory (denoted OOM).

MoE: We train a *mixture of experts* [18], where each *expert* is a separate Vanilla model trained on a data subset specified by the SF, $\lambda_i$. A gating network [37] is then trained to combine expert predictions into a final prediction.

HPS: In the style of multi-task learning, we model slices as separate task heads with a shared backbone trained via *hard parameter sharing*. Each slice task performs the same prediction task, but they are trained on subsets of data corresponding to $\lambda_i$. In this approach, backpropagation from different slice tasks is intended to encourage a slice-aware representation bias [7, 35].

Manual: To simulate the manual effort required to tune slice-specific hyperparameters, we leverage the same architecture as HPS and grid search over loss term multipliers, $\alpha \in \{2, 20, 50, 100\}$, for underperforming slices based on Vanilla model predictions (i.e. $score_{overall} - score_{slice} \geq 5$ F1).

### 4.1.2 Datasets

**NLU Datasets.** We select slices based on independently-conducted error analyses [20] (Appendix **??**). In **Corpus of Linguistic Acceptability (CoLA)** [40], the task is to predict whether a sentence is linguistically acceptable (i.e. grammatically); we measure performance using the Matthews correlation coefficient [24]. Natural slices might occur as questions or long sentences, as corresponding examples might consist of non-standard or challenging sentence structure. Since ground truth test labels are not available for this task (they are held out in evaluation servers [39]), we sample to create data splits with 7.2K/1.3K/1K train/valid/test sentences, respectively. To properly evaluate slices of interest, we ensure that the proportions of examples in ground truth slices are consistent across splits. In **Recognizing Textual Entailment (RTE)** [3, 4, 10, 15, 39], the task is to predict whether or not a premise sentence entails a hypothesis sentence. Similar to CoLA, we create our own data splits and use 2.25K/0.25K/0.275K train/valid/test sentences, respectively. Finally, in a user study where we work with practitioners tackling the **SuperGlue** [38] benchmark, we leverage *Slice-based Learning* to improve state-of-the-art model quality on benchmark submissions.

**CV Dataset.** In the image domain, we evaluate on an autonomous vehicle dataset called **Cyclist Detection for Autonomous Vehicles (CyDet)** [22]. We leverage clips in a self-driving video dataset to detect whether a cyclist (person plus bicycle) is present at each frame. We select one independent clip for evaluation, and the remainder for training; for valid/test splits, we select alternating batches of five frames each from the evaluation clip. We preprocess the dataset with an open-source implementation of Mask R-CNN [23] to provide metadata (e.g. presence of traffic lights, benches), which serve as slice indicators for each frame.

### 4.1.3 Results

**Slice-aware models improve slice-specific performance.** We see in Table 1 that each slice-aware model (HPS, Manual, MoE, SBL) largely improves over the naive model.

**SBL improves overall performance.** We also observe that SBL improves overall performance for each of the datasets. This is likely because the chosen slices were explicitly modeled from error analysis papers, and explicitly modeling "error" slices led to improved overall performance.

**SBL learns slice expert representations consistently.** While HPS and Manual perform well on some slices, they exhibit much higher variance compared to SBL and MoE (as denoted by the

std. in Table 1). These baselines lack an attention mechanism to reweight slice representations in a consistent way; instead, they rely purely on representation bias from slice-specific heads to improve slice-level performance. Because these representations are not modeled explicitly, improvements are largely driven by chance, and this approach risks worsening performance on other slices or overall.

**SBL improves performance with a parameter-efficient representation.** For **CoLA** and **RTE** experiments, we used the `BERT-base` [13] architecture with 110M parameters; for **CyDet**, we used `ResNet-18` [16]. For each additional slice, SBL requires a 7% and 5% increase in relative parameter count in the BERT and ResNet architectures, respectively (total relative parameter increase reported in Table 1). As a comparison, HPS requires the same relative increase in parameters per slice. MOE on the other hand, increases relative number of parameters by 100% per slice for both architectures. With limited increase in model size, SBL outperforms or matches all other baselines, including MOE, which requires an order of magnitude more parameters.

**SBL improves state-of-the-art quality models with slice-aware representations.** In a submission to SuperGLUE evaluation servers, we leverage the same `BERT-large` architecture as previous submissions and observe improvements on NLU tasks: +3.8/+2.8 avg. F1/acc. on CB [12], +2.4 acc. on COPA [34], +2.5 acc. on WiC [28], and +2.7 on the aggregate benchmark score.

## 4.2 Weak Supervision Comparisons

To contextualize our contributions in the weak supervision literature, we compare directly to Data Programming (DP) [29], a popular approach for reweighting user-specified heuristics using supervision source information [31]. We consider two text-based relation extraction datasets: **Chemical-Disease Relations (CDR)**,[41], in which we identify causal links between chemical and disease entities in a dataset of PubMed abstracts, and **Spouses** [9], in which we identify mentions of spousal relationships using preprocessed pairs of person mentions from news articles (via Spacy [17]). In both datasets, we leverage the exact noisy linguistic patterns and distant supervision heuristics provided in the open-source implementation of DP. Rather than voting on a particular class, we repurpose the provided labeling functions as binary slice indicators for our model. We then train our slice-aware model on the probabilistic labels aggregated from these heuristics.

**SBL improves over current weak supervision methods.** Treating the noisy heuristics as slicing functions, we observe lifts of up to 1.3 F1 overall and 15.9 F1 on heuristically-defined slices. We reproduce the DP [29] setup to obtain overall scores of F1=41.9 on **Spouses** and F1=56.4 on **CDR**. Using *Slice-based Learning*, we improve to $42.8$ $(+0.9)$ and $57.7$ $(+1.3)$ F1, respectively. Intuitively, we can explain this improvement, because SBL has access to features of the data belonging to slices whereas DP relies only on the source information of each heuristic.

## 5 Conclusion

We introduced the challenge of improving slice-specific performance without damaging the overall model quality, and proposed the first programming abstraction and machine learning model to support these actions. We demonstrated that the model could be used to push the state-of-the-art quality. In our analysis, we can explain consistent gains in the *Slice-based Learning* paradigm, as our attention mechanism has access to a rich set of deep features, whereas existing weak supervision paradigms have no way to access this information. We view this work in the context of programming models that sit on top of traditional modeling approaches in machine learning systems.

*Acknowledgements* We would like to thank Braden Hancock, Feng Niu, and Charles Srisuwananukorn for many helpful discussions, tests, and collaborations throughout the development of slicing. We gratefully acknowledge the support of DARPA under Nos. FA87501720095 (D3M), FA86501827865 (SDH), FA86501827882 (ASED), NIH under No. U54EB020405 (Mobilize), NSF under Nos. CCF1763315 (Beyond Sparsity) and CCF1563078 (Volume to Velocity), ONR under No. N000141712266 (Unifying Weak Supervision), the Moore Foundation, NXP, Xilinx, LETI-CEA, Intel, Microsoft, NEC, Toshiba, TSMC, ARM, Hitachi, BASF, Accenture, Ericsson, Qualcomm, Analog Devices, the Okawa Foundation, and American Family Insurance, Google Cloud, Swiss Re, and members of the Stanford DAWN project: Teradata, Facebook, Google, Ant Financial, NEC, SAP, VMWare, and Infosys. The U.S. Government is authorized to reproduce and distribute reprints for Governmental purposes notwithstanding any copyright notation thereon. Any opinions, findings, and conclusions or recommendations expressed in this material are those of the authors and do not necessarily reflect the views, policies, or endorsements, either expressed or implied, of DARPA, NIH, ONR, or the U.S. Government.

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
