[Supplementary Material]

## A1 Appendix

### A1.1 Model Characteristics

We include summarized model characteristics and the associated baselines to supplement Sections **??** and **??**.

| Method | Slice-aware | No manual tuning | Weighted slice info. | Avoids copies of model ($M$ params) | Num. Params |
|--------|:-----------:|:----------------:|:--------------------:|:-----------------------------------:|:-----------:|
| VANILLA |  | ✓ |  | ✓ | $O(M + r)$ |
| HPS | ✓ | ✓ |  | ✓ | $O(M + kr)$ |
| MANUAL | ✓ |  | ✓ | ✓ | $O(M + kr)$ |
| MOE | ✓ | ✓ | ✓ |  | $O(kM + kr)$ |
| SBL | ✓ | ✓ | ✓ | ✓ | $O(M + krd')$ |

Table 1: **Model characterizations**: We characterize each model's advantages/limitations and compute the number of parameters for each baseline model, given $k$ slices, $M$ backbone parameters, feature representation $z$ dimension $r$, and slice expert representation $p_i$ dimension $d'$.

### A1.2 Slicing Function (SF) Construction

We walk through specific examples of SFs written for a number of our applications.

**Textual SFs** For text-based applications (COLA, RTE), we write SFs over pairs of sentences for each task. Following dataset convention, we denote the first sentence as the *premise* and the second as the *hypothesis* where appropriate. Then, SFs are written, drawing largely from existing error analysis [**?** ]. For instance, we might expect certain questions to be especially difficult to formulate in a language acceptability task. So, we write the following SF to heuristically target *where* questions:

```python
def SF_where_question(premise, hypothesis):
    # triggers if "where" appears in sentence
    sentences = premise + hypothesis
    return "where" in sentences.lower()
```

In some cases, we write SFs over both sentences at once. For instance, to capture possible errors in article references (e.g. *the Big Apple* vs *a big apple*), we specify a slice where multiple instances of the same article appear in provided sentences:

```python
def SF_has_multiple_articles(premise, hypothesis):
    # triggers if a sentence has more than one occurrence of the same article
    sentences = premise + hypothesis
    multiple_a = sum([int(x == "a") for x in sentences.split()]) > 1
    multiple_an = sum([int(x == "an") for x in sentences.split()]) > 1
    multiple_the = sum([int(x == "the") for x in sentences.split()]) > 1
    return multiple_a or multiple_an or multiple_the
```

**Image-based SFs** For computer vision applications, we leverage image metadata and bounding box attributes, generated from an off-the-shelf Mask R-CNN [**?** ], to target slices of interest.

```python
def SF_bus(image):
    # triggers if a "bus" appears in the predictions of the noisy detector
    outputs = noisy_detector(image)
    return "bus" in outputs
```

We note that these potentially expensive detectors are *non-servable*—they run offline, and our model uses learned indicators at inference time. Despite the detectors' noisy predictions, our model is able to to reweight representations appropriately.

### A1.3  CoLA SFs

CoLA is a language acceptability task based on linguistics and grammar for individual sentences. We draw from error analysis which introduces several linguistically imortance slices for language acceptability via a series of challenge tasks. Each task consists of synthetically generated examples to measure model evaluation on specific slices. We heuristically define SFs to target subsets of data corresponding to each challenge, and include the full set of SFs derived from each category of challenge tasks:

- **Wh-words**: This task targets sentences containing *who, what, where, when, why, how*. We exclude *why* and *how* below because the CoLA dataset does not have enough examples for proper training and evaluation of these slices.

```
def SF_where_in_sentence(sentence):
    return "where" in sentence

def SF_who_in_sentence(sentence):
    return "who" in sentence

def SF_what_in_sentence(sentence):
    return "what" in sentence

def SF_when_in_sentence(sentence):
    return "when" in sentence
```

- **Definite-Indefinite Articles**: This challenge measures the model based on different combinations of definite (*the*) and indefinite (*a,an*) articles in a sentence (i.e. swapping definite for indefinite articles and vice versa). We target containing multiple uses of a definite (*the*) or indefinite article (*a, an*):

```
def SF_has_multiple_articles(sentence):
    # triggers if a sentence has more than one occurrence of the same article
    multiple_indefinite = sum([int(x == "a") for x in sentence.split()]) > 1 or sum([
     int(x == "an") for x in sentence.split()]) > 1
    multiple_definite = sum([int(x == "the") for x in sentence.split()]) > 1

    return multiple_indefinite or multiple_definite
```

- **Coordinating Conjunctions**: This task seeks to measure correct usage of coordinating conjunctions (*and, but, or*) in context. We target the presence of these words in both sentences.

```
def and_in_sentence(sentence):
    return "and" in sentence

def but_in_sentence(sentence):
    return "but" in sentence

def or_in_sentence(sentence):
    return "or" in sentence
```

- **End-of-Sentence**: This challenge task measures a model's ability to identify coherent sentences or sentence chunks after removing puctuation. We heuristically target this slice by identifying particularly short sentences and those that end with verbs and adverbs. We use off-the-shelf parsers (i.e. Spacy [?]) to generate part-of-speech tags.

```
def SF_short_sentence(sentence):
    # triggered if sentence has fewer than 5 tokens
    return len(sentence.split()) < 5

# Spacy tagger
def get_spacy_pos(sentence):
  import spacy
  nlp = spacy.load("en_core_web_sm")
  return nlp(sentence).pos_

def SF_ends_with_verb(sentence):
    # remove last token, which is always punctuation
    sentence = sentence[:-1]
    return get_spacy_pos(sentence)[-1] == "VERB"

def SF_ends_with_adverb(sentence):
    # remove last token, which is always punctuation
```

```
    sentence = sentence[:-1]
    return get_spacy_pos(sentence)[-1] == "ADVERB"
```

## A1.4    RTE SFs

Similar to CoLA, we use challenge tasks from NLI-based error analysis [? ] to write SFs over the textual entailment (RTE) dataset.

- **Prepositions**: In one challenge, the authors swap prepositions in the dataset with prepositions in a manually-curated list. The list in its entirety spans a large proportion of the RTE dataset, which would constitute a very large slice. We find it more effective to separate these prepositions into *temporal* and *possessive* slices.

```
def SF_has_temporal_preposition(premise, hypothesis):
    temporal_prepositions = ["after", "before", "past"]
    sentence = premise + sentence
    return any([p in sentence for p in temporal_prepositions])

def SF_has_possessive_preposition(premise, hypothesis):
    possessive_prepositions = ["inside of", "with", "within"]
    sentence = premise + sentence
    return any([p in sentence for p in possessive_prepositions])
```

- **Comparatives**: One challenge chooses sentences with specific comparative words and mutates/negates them. We directly target keywords identified in their approach.

```
def SF_is_comparative(premise, hypothesis):
    comparative_words = ["more", "less", "better", "worse", "bigger", "smaller"]
    sentence = premise + hypothesis
    return any([p in sentence for p in comparative_words])
```

- **Quantification**: One challenge tests natural language understanding with common quantifiers. We target common quantifiers in both the combined premise/hypothesis and in *only* the hypothesis.

```
def is_quantification(premise, hypothesis):
    quantifiers = ["all", "some", "none"]
    sentence = premise + hypothesis
    return any([p in sentence for p in quantifiers])

def is_quantification_hypothesis(premise, hypothesis):
    quantifiers = ["all", "some", "none"]
    return any([p in hypothesis for p in quantifiers])
```

- **Spatial Expressions**: This challenge identifies spatial relations between entities (i.e. *A is to the left of B*). We exclude this task from our slices, because such slices do not account for enough examples in the *RTE* dataset.

- **Negation**: This challenge task identifies whether natural language inference models can handle negations. We heuristically target this slice via a list of common negation words from a top result in a web search.

```
def SF_common_negation(premise, hypothesis):
    # Words from https://www.grammarly.com/blog/negatives/
    negation_words = [
        "no",
        "not",
        "none",
        "no one",
        "nobody",
        "nothing",
        "neither",
        "nowhere",
        "never",
        "hardly",
        "scarcely",
        "barely",
        "doesnt",
        "isnt",
        "wasnt",
        "shouldnt",
        "wouldnt",
        "couldnt",
```

```
        "wont",
        "cant",
        "dont",
    ]
    sentence = premise + hypothesis
    return any([x in negation_words for x in sentence])
```

- **Premise/Hypothesis Length**: Finally, separate from the cited error analysis, we target different length hypotheses and premises as an additional set of slicing tasks. In our own error analysis of the RTE model, we found these represented intuitive slices: long premises are typically harder to parse for key information, and shorter hypotheses tend to share syntactical structure.

```
def SF_short_hypothesis(premise, hypothesis):
    return len(hypothesis.split()) < 5

def SF_long_hypothesis(premise, hypothesis):
    return len(hypothesis.split()) > 100

def SF_short_premise(premise, hypothesis):
    return len(premise.split()) < 15

def SF_long_premise(premise, hypothesis):
    return len(premise.split()) > 100
```

## A1.5   CYDET SFs

For the cyclist detection dataset, we identify subsets that correspond to other objects in the scene using a noisy detector (i.e. an off-the-shelf Mask R-CNN [? ]).

```
# define noisy detector
def noise_detector(image):
  probs = mask_rcnn.forward(image)

  # threshold predictions
  preds = []
  for object in classes:
      if probs["object"] > 0.5:
          preds.append(object)
  return preds

# Cyclist Detection SFs
def SF_bench(image):
    outputs = noisy_detector(image)
    return "bench" in outputs

def SF_truck(image):
    outputs = noisy_detector(image)
    return "truck" in outputs

def SF_car(image):
    outputs = noisy_detector(image)
    return "car" in outputs

def SF_bus(image):
    outputs = noisy_detector(image)
    return "bus" in outputs

def SF_person(image):
    outputs = noisy_detector(image)
    return "person" in outputs

def SF_traffic_light(image):
    outputs = noisy_detector(image)
    return "traffic light" in outputs

def SF_fire_hydrant(image):
    outputs = noisy_detector(image)
    return "fire hydrant" in outputs

def SF_stop_sign(image):
    outputs = noisy_detector(image)
    return "stop sign" in outputs

def SF_bicycle(image):
    outputs = noisy_detector(image)
    return "bicycle" in outputs
```

### A1.6 Slice-specific Metrics

We visualize slice-specific metrics across each application dataset, for each method of comparison. We report the corresponding aggregate metrics in Figure 1 (below).

In COLA, we see that MOE and SBL exhibit the largest slice-specific gains, and also overfit on the same slice *ends with adverb*. In RTE, we see that SBL improves performance on all slices except *common negation*, where it falls less than a point below VANILLA. On CYDET, we see the largest gains for SBL on *bench* and *bus* slices—in particular, we are able to improve in cases where the model might able to use the presence of these objects to make more informed decisions about whether a cyclist is present. Note: because the MOE model on CYDET encounters an "Out of Memory" error, the corresponding (blue) data bar is not available for this dataset.

Figure 1: For each application dataset (Section 4.1) we report all relative, slice-level metrics compared to VANILLA for each model.