[Reviews · NeurIPS 2019]

Reviewer 1



Originality: - The authors claim this paradigm towards specifying ML models is novel. It is somewhat difficult for me to assess the originality of this work as it's not exactly my area, but I am inclined to agree that their approach seems new and interesting. Quality: - Section 3.3: It's quite generous to call these "key properties" of the model, as really they refer to the results of this particular instantiation of slice-based learning on this toy dataset. It's definitely nice to see that the approach works on a toy dataset, but I would strongly consider reframing this section. - The authors point out three challenges in building their slice-based learning framework: coping with noise (in the slice labeling functions), stable improvement of the model, scalability. The latter two are adequately addressed in the paper and experiments, but the noise aspect was not really addressed. It'd be nice to see an experiment where you add noise to the labeling functions to see how model performance varies in the amount of noise, as a lot of interesting slice labeling functions would likely be noisy in text and vision. Relatedly, is Figure 3c missing? - In the experiments, the proposed approach gets modest gains (some well within a standard deviation of other baselines), and in the appendix there are some counterintuitive trends on some of the slices where the propsed approach improves where other baselines hurt and vice versa. I don't think this is fatal to the paper as it's still interesting work and seems somewhat effective, but it'd be nice to have some explanation/speculation of why this is. Clarity: - The exposition of the framework in Section 3 is nice and very readable; the figure and lettered model components do a lot of work here. - Section 4.1: This section seems to be missing important details. (1) what do you mean by "we created our own data splits"? Did you collect new data or re-use existing labeled data? (2) What are the slices? This seems quite important and interesting to know, even as a brief, general statement (e.g., "We create slices for CoLA based on the presence of various wh- question words."). The information is in the appendix, but definitely should be in the main paper. - Section 4.2 and generally: It would really be nice to have slightly more informative descriptions of the baselines, e.g. what is data-programming? (This might just be me not being that familiar with this area, but if it is an "emerging class of programming models", it would seem like not that many people know about it and worth explaining!) - Section 4.3: Related to the above, it is very odd not to show performance per slice in the main table. I know it's in the appendix, but a main research question is improving performance on the slice w/o hurting performance overall, and the table as presented does not show that. There's also two entire experiments contained in S4.3, lines 283-292; that is not enough time or space for readers to get a since of what the experiment is. - L214: "numebr" Significance: Overall, this work feels quite significant to me, though again, it is somewhat orthogonal to my area. The proposed approach does seem competitive with the "state-of-the-art" (mixture of experts) while being significantly less resource intensive. I believe that industry ML practitioners will likely use this approach and other academic groups in this area will take inspiration from the ideas and try to build on them.

Reviewer 2



The paper introduces a novel slice-based programming abstraction. It presents a method for training models given the slice information that is closely related to past work on ensembles, mixture-of-experts, and multi-task learning. The main advantage for the proposed method is that it limits the number of model parameters required; otherwise it is very similar to instantiating a mixture-of-exports model on top of shared backbone features (rather than the inputs directly). The slice-based abstraction introduced is highly applicable to practical applications of ML model, and has the potential to be widely used for ML deployments. The presentation of the model architecture (Sec 3.2 and Figure 2) is not very clear, so I haven't been able to fully figure out the details of the approach. As I understand it at a high level, the architecture involves computing slice-specific representations, and then combines them into a single representation based on a module that predicts which slices are active. However, the text confused me more than it helped me understand the fine details of the approach. It might be helpful to refine the presentation, and also update Figure 2 to use distinct visual cues when presenting quantities such as attention logits, attention probabilities, and ground-truth slice membership labels as computed by the slicing functions. [Thank you for promising to clarify the presentation here; I look forward to seeing a revised version of this section. The author response points out the text assumed a binary classification setting where there was only one logit, instead of one per class. This wasn't clear to me so I kept expecting there to be vectors of class-specific logits instead.] 160: the letter "p" is often used for probability distributions, but is used for hidden vectors here. Switching to another letter would have reduced my confusion in reading this section. 165: is g(P) introduced here the same as in section (d) previously? But in that case, g(p_i) in R^c from the previous section doesn't dimension match with g(P) in R^(k+1) on line 165. Maybe g(P) is in R^(k+1 x c) instead, or perhaps I'm misunderstanding? 166: What does "abs" here refer to? It looks like you're reducing a set of logits to a scalar confidence score. My only thought is component-wise absolute value, but that can't be right. 173.5: I don't quite follow how the model makes its final prediction. Earlier in the paper the authors claim that their method "model[s] residuals between slice-level and the overall task" (56-57), whereas it seems here that the original backbone representation "z" is no longer used here (except as part of p_BASE earlier). Also, what is the motivation for doing the slice-based modeling using dimensionality "h", rather than the dimensionality of the backbone (r)? It sounds like the backbone features are being projected down to a lower-dimensional space; is there a concern that this will discard relevant features? 56: attenton -> attention Table 1: +/- -2.1 should be positive 2.1

Reviewer 3



Update: I read authors' response and I am satisfied with it. I like the plot that they included, which shows that indicator becomes less certain the more noise is encountered. I would suggest to include it into paper's final version, space permitting. I also find architecture impact responce interesting, and again suggest to try to put it into the main paper. I keep my Accept rating Summary: this paper operates in a setting when it is known that the model must perform well on a some subsets (not nessesarily not overlapping) of data. The authors formulate this problem as multiple objectives problem (lernt generic representation, learn slice representation, combine with self attention) and demonstrate in a number of experiments that the method is able to both improve performance on the slices and overall performance of the models Detailed review: Overall an interesting paper with several good observations. I like the learnt "indicators", that account for noise in slices. I assume it also allows for the slices that developers "think " are important but might turn out to be not. Concerns: - The architecture seems a bit of black magic. For example, are all parts required and crutial? Is it designed or "found" during extensive search for architectures? - The combined loss (sum of the sublosses) - why all the objectives are equally important (weight 1). Did you try to introduce hyperparams there that pay more attention, for example, to l_pred - what would a dummy baseline that introduces losses on each slices and then sums it with the normal weight (may be with some weights) learn? Basically it is a continuation to a question of which part is the most important - experiments: for figure1 - it seems with the current setup u are fitting to noise. What would be more indicative is to introduce also a slice that is noise but not critical (eg have 3 slices, 2 are included in your learning, 3rd was used to generate the data but not included - eg no slice indicator func) and see what it does on the third non critical slice - Is the idea that learnt indicator allows to understand what slice is noisy? Because how does a developer knows that the slice is important or it is just a more noisy segment of the data, for example where the measurements for the car are not as precise (night vs day speed detection) Questions: - how many slices are you able to handle? From your experience, did you observe that it is beneficial to include all potential slices or it hinders the model - what happens when you don't learn the "indicators" and just use hard indicator functions (both during training and inference)? I would assume it would still improve on slices, but would be more susceptible to noise Minor line 75 this procedure yields achieves -> choose one

[Author Response · NeurIPS 2019]

We thank the reviewers for their time and thoughtful feedback. Taking their helpful comments into account, we sought
and received additional feedback to extend and clarify the presentation of our work. Since submission, we have also
achieved **state-of-the-art scores** on the SuperGLUE benchmark by 2.7 points using the proposed approach.

**Ablation studies. [R1, R2, R3]** We thank all reviewers for their
detailed feedback about the proposed architecture. We have added
ablation studies to Section 3.3 to clarify specific components and
will include additional experiments and details in the final draft.
**Coping with noise**: We test the robustness of our approach on
a simple synthetic example: in the Figure to the right, we show
noisy SFs (top row: no noise, 40% noise, 80% noise) and the
corresponding slice indicator's output as a heatmap (bottom row:
darker indicates higher likelihood of slice-membership). We show
that the indicator assigns low relative probabilities on noisy (40%)
SF samples (bottom middle) and ignores a very noisy (80%) SF,
assigning relatively uniform scores to all samples (bottom right).

**Architecture ablations**: We thank R2 and R3 for suggestions to clarify the contributions of the architecture's compo-
nents. We perform an ablation study using a synthetic, binary classification dataset with four slices covering random
data subsets. We observe that indicator outputs contribute +3.4 F1; without this indicator module, the model might fail
to handle noisy SFs. The predictor confidences contribute +4.6 F1; without considering these confidences, the attention
mechanism might combine non-expert features into the reweighted representation. Compared to equal weights, our
attention mechanism contributes +5.6 F1; without attention, there is no fine-grained combination of slice representations.

**Presentation of model architecture. [R2, R3]** In response to R2's feedback, we have updated each module with
dimension annotations and updated Figure 2 with visual cues to specify where slice labels are used during training (i.e.
as *labels* for training indicators and *masks* for training predictors). Following R3's suggestion, we have more clearly
expressed the costs (e.g. model size, training time) of each approach, especially MoE, with respect to OURS in Table 1.

We thank R2 for pointing out vague notation in Section 3.2, which we have clarified in the updated draft. In L160,
we changed $p$ to $w$ to avoid confusion with probability notations. In L165, $g(P)$ (from Section 3.2(e)) indeed refers
to the concatenation of each $g(p_i)$ from Section 3.2(d). Our notation refers to the binary setting, where $c = 1$ such
that $P \in \mathbb{R}^{h \times k+1}$. In L166, we clarify that the predictor confidence is computed using the maximum probability over
prediction classes. Our experiments are binary classification tasks, for which we use the absolute value of the predictor
logit as this confidence. Additionally, R2 is correct that $z$ is not used in these reweighted features, $z'$. Instead, the base
representation is modeled as a trivial slice: $p_{BASE}$, and a final prediction is made based on the reweighted $z'$. We
clarify that $h$ is a hyperparameter for flexibility in specific applications; for simplicity, we set this to $r$ in experiments.

**R1:** We agree with R1 that Section 3.3 could be reframed more conservatively. We have updated the title to "Synthetic
Experiments" to clarify that our observations are grounded in the synthetic setting. R1 asked about a missing Figure 3c;
Figure 3c refers to the right-most graphic in Figure 3, and we have labeled this more explicitly in the updated draft.

R1 asked for clarification about experimental details and provided feedback for the organization of Section 4. In
our data splits, we ensure that the proportion of examples belonging to each slice is equivalent across train/valid/test
for appropriate evaluation of slices; we did not collect/re-use different data sources. Furthermore, we moved SF
implementation/evaluation details and a description of slices from the appendix to the body of the paper. In Section 4.3,
we have included an error analysis regarding counter-intuitive trends in slice-specific performance. For example, results
on COLA may be explained by limitations in the backbone architecture; we observe low performance on MoE, which
has extra capacity, on certain slices (e.g. *ends with adverb* and *has but*) where OURS also underperforms. Following
R1's suggestion, we have included detailed descriptions about the baselines. Specifically, we anchor our work in data
programming [23], a baseline from weak supervision literature that learns to combine noisy, user-provided heuristics.

**R2:** We thank R2 for feedback on our empirical evaluation. In experiments, we use strong baselines as backbones:
pre-trained BERT++ (proposed by SuperGLUE organizers) for COLA/RTE and pre-trained ResNet for CYDET. We
thank R2 for the suggestion to report the evaluation server results; we obtain three new SOTA scores on SuperGLUE
tasks: (+3.8/+2.8 Avg. F1/acc. on CB, +2.8 acc. on COPA, +2.5 acc. on WiC).

**R3:** R3 asked about the weighting of our loss terms. In practice, one may set a hyperparameter for each loss term; to
simplify our study, we set all weights to 1. R3 is also correct that "hard" slice features would be more susceptible to
noise; we will include this baseline in Table 1. We emphasize, however, that introducing such features would violate the
key assumption that slice information/metadata is not available during inference, as discussed in Section 3.1. Finally,
R3 asked how many slices our model can support. Our experiments include an average of 10 slices per application,
while in an industrial collaboration, we have deployed the *Slice-based Learning* on hundreds of production slices.

[Meta-Review · NeurIPS 2019]

Post rebuttal, all reviewers agree this paper should be accepted. A clear accept.